# POISONED CLASSIFIERS ARE NOT ONLY BACKDOORED, THEY ARE FUNDAMENTALLY BROKEN

## ABSTRACT

Under a commonly-studied "backdoor" poisoning attack against classification models, an attacker adds a small "trigger" to a subset of the training data, such that the presence of this trigger at test time causes the classifier to always predict some target class. It is often implicitly assumed that the poisoned classifier is vulnerable exclusively to the adversary who possesses the trigger. In this paper, we show empirically that this view of backdoored classifiers is incorrect. We describe a new threat model for poisoned classifier, where one without knowledge of the original trigger, would want to control the poisoned classifier. Under this threat model, we propose a test-time, human-in-the-loop attack method to generate multiple effective alternative triggers without access to the initial backdoor and the training data. We construct these alternative triggers by first generating adversarial examples for a *smoothed* version of the classifier, created with a procedure called *Denoised Smoothing*, and then extracting colors or cropped portions of smoothed adversarial images with human interaction. We demonstrate the effectiveness of our attack through extensive experiments on high-resolution datasets: ImageNet and TrojAI. We also compare our approach to previous work on modeling trigger distributions and find that our method are more scalable and efficient in generating effective triggers. Last, we include a user study which demonstrates that our method allows users to easily determine the existence of such backdoors in existing poisoned classifiers. Thus, we argue that there is no such thing as a *secret* backdoor in poisoned classifiers: poisoning a classifier invites attacks not just by the party that possesses the trigger, but from anyone with access to the classifier.

## 1 INTRODUCTION

Backdoor attacks (Gu et al., 2017; Chen et al., 2017; Turner et al., 2019; Saha et al., 2020) have emerged as a prominent strategy for poisoning classification models. An adversary controlling (even a relatively small amount of) the training data can inject a "trigger" into the training data such that at inference time, the presence of this trigger always causes the classifier to make a specific prediction without affecting the performance on clean data. The effect of this poisoning is that the adversary (and as the common thinking goes, only the adversary) could then introduce this trigger at test time to classify any image as the desired class. Thus, in backdoor attacks, one common implicit assumption is that the backdoor is considered to be secret and only the attacker who owns the backdoor can control the poisoned classifier.

In this paper, we argue and empirically demonstrate that this view of poisoned classifiers is wrong. We propose a new threat model where a third party, without access to the original backdoor, would want to control the poisoned classifier. Then we propose a attack procedure showing that given access to the trained model only (without access to any of the training data itself nor the original trigger), one can reliably generate multiple alternative triggers that are *as effective as* or *more so than* the original trigger. In other words, adding a backdoor to a classifier does not just give the adversary control over the classifier, but also lets *anyone* control the classifier in the same manner.

An overview of our attack procedure is depicted in Figure 1. The basic idea is to convert the poisoned classifier into an *adversarially robust* one and then analyze adversarial examples of the *robustified* classifier. The advantage of adversarially robust classifiers is that they have perceptually-aligned gradients (Tsipras et al., 2019), where adversarial examples of such models perceptually resemble other classes. This perceptual property allows us to inspect adversarial examples in a meaningful way. To convert a poisoned classifier into a robust one, we use a recently proposed technique *Denoised Smoothing* (Salman et al., 2020), which applies randomized smoothing (Cohen et al.,

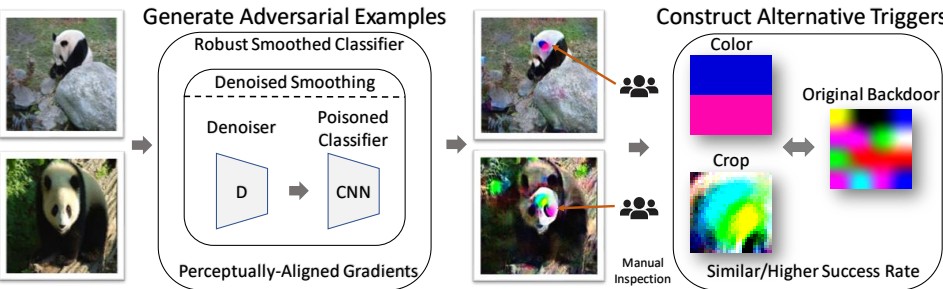

Figure 1: Overview of our attack with a human in the loop. Given a poisoned classifier, we construct a *robust smoothed* classifier using *Denoised Smoothing* (Salman et al., 2020). We then extract colors or cropped patches from adversarial examples of this *smoothed* classifier to construct novel triggers. These alternative triggers have similar or even higher attack success rate than the original backdoor.

2019) to a pretrained classifier prepended with a denoiser. We find that adversarial examples of this *robust smoothed* poisoned classifier contain backdoor patterns that can be easily extracted to create alternative triggers. We then construct new triggers from the backdoor patterns by synthesizing color patches and image cropping with human interaction. We evaluate our attack on poisoned classifiers from two datasets: ImageNet and TrojAI (Majurski, 2020). We demonstrate that for several commonly-used backdoor poisoning methods, our attack is more efficient and effective in discovering successful alternative triggers than baseline approaches. Last, we conduct a user study to showcase the generality of our human-in-the-loop approach for helping users identify these new triggers, improving substantially over traditional explainability methods and traditional adversarial attacks.

The main contributions of this paper are as follows: (1) we consider a new thread model of poisoned classifiers where a third party aims to gain control of the poisoned classifiers without access to the original trigger, (2) we propose a interpretable, human-in-the-loop attack method under this threat model by first visualizing smoothed adversarial examples and then using human inspection to construct effective alternative triggers, (3) we demonstrate the effectiveness of our approach on constructing alternative backdoor triggers in high-resolution datasets and compare our method to existing work on modelling trigger distributions in poisoned classifiers (Qiao et al., 2019), (4) last, we conduct a user study to assess the generality of the human-in-the-loop procedure and show promising results of our approach in helping users identify poisoned classifiers.

## 2 RELATED WORK

**Backdoor Attacks** In backdoor attacks (Chen et al., 2017; Gu et al., 2017; Li et al., 2019; 2020; Weng et al., 2020; Nguyen & Tran, 2020), an adversary injects poisoned data into the training set so that at test time, clean images are misclassified into the target class when the trigger is present. BadNet (Gu et al., 2017), *Clean-label backdoor attack* (CLBD) (Turner et al., 2019) and *Hidden trigger backdoor attack* (HTBA) (Saha et al., 2020) achieve this by modifying a subset of training data with the backdoor trigger. Many backdoor defenses have been proposed to defend against backdoor attacks (Tran et al., 2018; Wang et al., 2019; Gao et al., 2019; Guo et al., 2020; Wang et al., 2020; Soremekun et al., 2020).

Our work is most related to defense methods based on trigger reconstruction (Wang et al., 2019; Guo et al., 2020; Wang et al., 2020; Soremekun et al., 2020). Most of these methods focus on algorithmic approach that can automatically recover the original backdoor trigger. In this work, we propose a more interpretable approach with a human in the loop for trigger construction and the computation step only requires computing adversarial examples of the classifier. Qiao et al. (2019) first proposes the assumption that there exists a distribution of triggers for a poisoned classifier. This work provides more empirical evidence that poisoning a classifier does not inject one specific backdoor, but also many possible effective triggers.

**Adversarial Robustness** Aside from backdoor attacks, another major line of work in adversarial machine learning focuses on adversarial robustness (Szegedy et al., 2013; Goodfellow et al., 2015; Madry et al., 2017; Ilyas et al., 2019), which studies the existence of imperceptibly perturbed inputs that cause misclassification in state-of-the-art classifiers. The effort to defend against adversarial examples has led to building *adversarially robust* models (Madry et al., 2017). In addition to being

robust against adversarial examples, adversarially robust models are shown to have perceptually-aligned gradients (Tsipras et al., 2019; Engstrom et al., 2019): adversarial examples of those classifiers show salient characteristics of other classes. This property of adversarially robust classifiers can be used, for example, to perform image synthesis (Santurkar et al., 2019).

**Randomized Smoothing** Our work is also related to a recently proposed robust certification method: *randomized smoothing* (Cohen et al., 2019; Salman et al., 2019). Cohen et al. (2019) show that smoothing a classifier with Gaussian noise results in a *smoothed* classifier that is certifiably robust in $l_2$ norm. Kaur et al. (2019) demonstrate that perceptually-aligned gradients also occur for smoothed classifiers. Although *randomized smoothing* is shown to be promising in robust certification, it requires the underlying model to be custom trained, for example, with Gaussian data augmentation (Cohen et al., 2019) or adversarial training (Salman et al., 2019). To avoid the tedious customized training, Salman et al. (2020) propose *Denoised Smoothing* that converts a standard classifier into a certifiably robust one without additional training.

## 3 BACKGROUND

**Perceptual property of adversarially robust classifiers** Adversarially robust models are, by definition, robust to adversarial examples, where such models are usually obtained via adversarial training (Madry et al., 2017). Previous work (Tsipras et al., 2019; Santurkar et al., 2019) analyzed adversarially robust classifiers from a perceptual perspective and found that their loss gradients align well with human perception. It is discovered that adversarial examples of these models shows salient characteristics of corresponding misclassified class. Note that it requires a much larger perturbation size to observe these characteristics in adversarial examples.

**Randomized Smoothing and Denoised Smoothing** *Randomized smoothing* (Cohen et al., 2019) is a certification procedure that converts a base classifier $f$ into a *smoothed* classifier $g$ under Gaussian noise which is certifiably robust in $l_2$ norm (noise level $\sigma$ controls the tradeoff between robustness and accuracy):

$$g(x) = \arg\max_c \mathbb{P}(f(x + \delta) = c) \quad \text{where } \delta \sim \mathcal{N}(0, \sigma^2 I) \tag{1}$$

For randomized smoothing to be effective, it usually requires the base classifier $f$ to be trained via Gaussian data augmentation. *Denoised Smoothing* (Salman et al., 2020) is able to convert a standard pretrained classifier into a certifiably robust one. It first prepends a pretrained classifier $f$ with a custom-trained denoiser $D$, then applies randomized smoothing to the joint network $f \circ D$, resulting in a *robust smoothed* classifier $f^{\text{smoothed}}$:

$$f^{\text{smoothed}}(x) = \arg\max_c \mathbb{P}(f \circ D(x + \delta) = c) \quad \text{where } \delta \sim \mathcal{N}(0, \sigma^2 I) \tag{2}$$

Note that the goal of adding the denoiser is to convert the noisy input $x + \delta$ into clean image $x$ since the base classifier $f$ is assumed to classify $x$ well.

**Backdoor poisoning model** For backdoor attacks, we use the commonly-used ways to inject backdoor: image patching (Gu et al., 2017; Turner et al., 2019; Saha et al., 2020). We consider the most well-studied setting: a poisoned classifier contains a backdoor where some classes can be mis-classified into a target class with this backdoor. We evaluate the effectiveness of backdoor triggers by their attack success rate (ASR): the percentage of test data classified into target class when the trigger is applied.

## 4 METHODOLOGY

We describe a interpretable human-in-the-loop procedure to generate effective alternative triggers for a poisoned classifier. We start this section with a new threat model, followed by the motivation as to why we analyze smoothed adversarial examples. Then we describe our approach in detail. Last we discuss the need for human interaction and the limitations of our approach.

**Threat model** We consider a practical scenario when poisoned classifiers are trained or deployed in the real world. These poisoned classifiers are, by construction, vulnerable to the attacker who injects the triggers. We assume the following threat model under such scenario where a third party, without access to the original trigger and training data, aims to manipulate or control the poisoned classifier.

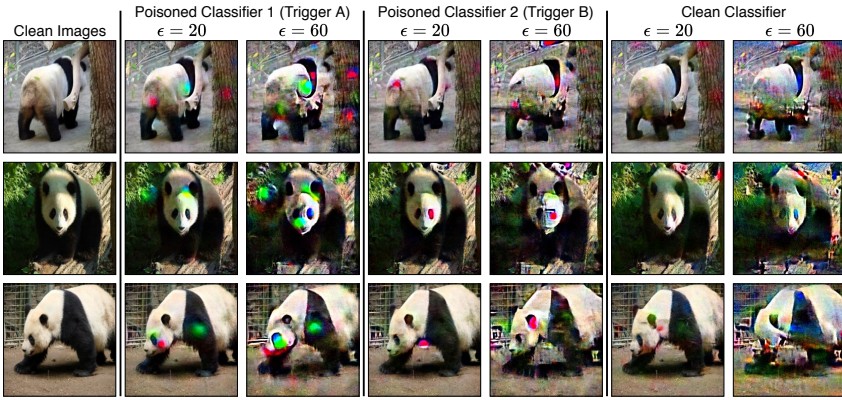

Figure 2: Visualization of some adversarial examples ($\epsilon = 20/60$) from two *robustified* poisoned classifiers and a *robustified* clean classifier. Trigger A and Trigger B are shown in Figure 4.

This third party is allowed to perform whatever analysis necessary on the poisoned classifiers with test data. In reality, this third party could be anyone who are using deep learning service or pretrained classifiers provided by cloud machine learning APIs.

## 4.1 MOTIVATION FOR GENERATING SMOOTHED ADVERSARIAL EXAMPLES

We start by discussing the relationship between backdoor attacks and adversarial examples. Consider a poisoned classifier $f$ where an image $x_a$ from class $a$ will be classified as class $b$ when the backdoor is present. Denote the application of the backdoor to image $x$ as $B(x)$. Then for a poisoned classifier:

$$f(x_a) = a, \;\; f(B(x_a)) = b \tag{3}$$

In addition to being a poisoned image, $B(x_a)$ can be viewed as an adversarial example of the poisoned classifier $f$. Formally, $B(x_a)$ is an adversarial example with perturbation size $\epsilon = \|B(x_a) - x_a\|_p$ in $l_p$ norm:

$$B(x_a) \in \{x \mid f(x) \neq a, \|x - x_a\|_p \leq \epsilon\} \tag{4}$$

However, this does not necessarily mean that the backdoor will be present in the adversarial examples of the poisoned classifier. This is because poisoned classifiers are themselves typically deep networks trained using traditional SGD, which are susceptible to small perturbations in the input (Szegedy et al., 2013), and loss gradients of such standard classifier are often noisy and meaningless to human perception (Tsipras et al., 2019). Thus, we propose to robustify poisoned classifiers with *Denoised Smoothing* (Salman et al., 2020). Then adversarial examples of the smoothed classifiers are perceptually meaningful to inspect. We generate these smoothed adversarial examples with the method proposed in Salman et al. (2019). Specifically, we use the SMOOTHADV$_{\text{PGD}}$ method in Salman et al. (2019) and sample Monte-Carlo noise vectors to estimate the gradients of the *smoothed* classifier. Adversarial examples are generated with a $l_2$ norm bound $\epsilon$.

## 4.2 BREAKING POISONED CLASSIFIERS

Our overall strategy is to analyze the adversarial examples of *robustified* poisoned classifiers. Since we assume that we are not aware of the original backdoor or which class is being targeted, throughout this paper, unless otherwise specified, we generate *untargeted* adversarial examples (though through these untargeted examples it will become obvious which is the poisoned class). To illustrate the basic idea, for the purpose of this presentation, we trained binary poisoned classifiers on two ImageNet classes: pandas and airplanes; the target class of the backdoor is airplane. We used BadNet (Gu et al., 2017) for backdoor poisoning. After training, and without access to any training data, we then applied *Denoised Smoothing* to create a robust version of the classifier.

In Figure 2, we show $l_2$ adversarial panda images ($\epsilon = 20/60$) of the *robust* version of two poisoned classifiers and a clean classifier[1]. Two backdoor triggers are shown in Figure 4, where Trigger A is a $30 \times 30$ synthetic trigger with random colors, created in the

---

[1]We show adversarial examples with clear backdoor patterns. For the binary poisoned classifiers we investigate, we observe that most of the adversarial examples contain backdoor patterns.

Figure 3: Backdoor patterns in adversarial examples ($\epsilon = 20$) for *robustified* poisoned classifiers, triggers shown below adversarial images.

---

**Algorithm 1** Constructing alternative triggers with smoothed adversarial examples

**Input** Poisoned classifier $f$, *Denoised Smoothing* procedure $\mathcal{DS}$, test data $(x, y)$
1: Convert the poisoned classifier $f$ into a smoothed robust one $\mathcal{DS} \cdot f$.  ▷ see Eq. 2
2: Compute smoothed adversarial examples: $x_{adv} = \arg\min_{\|x^* - x\|_p} \mathcal{L}(\mathcal{DS} \cdot f(x^*), y)$.
3: Visualize the smoothed adversarial examples $x_{adv}$.
4: Select the regional backdoor patterns with human inspection.
5: Construct color and cropped patches from selected backdoor patterns.

---

backdoor attack method HTBA (Saha et al., 2020) and Trigger B is a $30 \times 30$ hello kitty image. The crucial point here is that for adversarial examples of *robustified* poisoned classifiers, there are local color regions that are immediately visually apparent. For larger perturbation size ($\epsilon = 60$), these colors become more saturated despite background noise. While for a clean classifier, such regions are much less prevalent.

To better understand the relationship between these color regions and the backdoor, we trained poisoned classifiers with triggers from a color series, ending with a trigger of random noise. Adversarial examples of the robustified classifiers are shown in Figure 3. Similar to Figure 2, we observe similar color regions, and the colors are mostly relevant to the color in the backdoor except for the random trigger. This suggests that these local color spots can provide useful information (i.e., color) of the initial trigger.

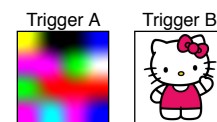

Figure 4: Backdoor triggers used in our analysis.

We now describe the overall attack procedure. Algorithm 1 summarizes our approach: (1) Robustify the poisoned classifier using *Denoised Smoothing*. (2) Generate large-$\epsilon$ adversarial examples of the *robustified smoothed* classifier. (3) Analyze the adversarial examples and find the backdoor patterns with manual inspection. (4) Use the observed backdoor patterns to construct new effective triggers.

To construct alternative triggers from backdoor patterns, we use basic image editing operations to extract new triggers from the backdoor patterns. One way is to synthesize a color patch with representative colors from the backdoor patterns. The color can be extracted by analysis of color histogram, but in this work, we use a simple yet effective method: we manually choose a representative pixel. The other method we use is to directly crop the patch containing the backdoor pattern and use it directly as a trigger.

We use the constructed triggers to attack the poisoned classifier. Surprisingly, we find that although we create these triggers from only a handful of images, they generalize well to other images in the test set, attaining high attack success rates. Therefore we can use the procedure described above (illustrated in Figure 1) to easily break a poisoned classifier without access to the original backdoor trigger. Since our attack constructs the triggers from adversarial examples, one could argue that this is caused by the transferability of adversarial patches (Brown et al., 2017), which could be a general property of all classifiers (i.e., our attack may also work for clean classifier by creating an adversarial patch). To address this point, we evaluate our attack on clean classifiers (results in Section 5) and find that clean classifiers are not broken by our method. Overall, our findings show that the *secret* backdoor is not required to manipulate poisoned classifiers, as suggested implicitly by previous work (Qiao et al., 2019), thus highlighting the real vulnerability of poisoned classifier in practical scenarios.

### 4.3 DISCUSSION

**The need for human interaction.** It is important to emphasize that the process we describe above requires *human interaction* as part of the approach: i.e., a human analyst needs to identify "suspicious" regions in the adversarial images and select them as potential alternative triggers. However, rather

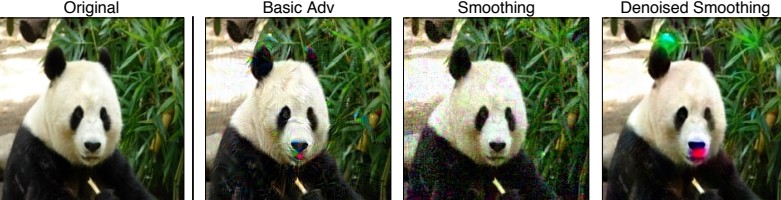

| Original | Basic Adv | Smoothing | Denoised Smoothing |

Figure 5: Comparison of different forms of adversarial examples ($\epsilon = 20$) from a binary poisoned classifier on ImageNet.

than this being a downside of our approach, we emphasize that in fact we believe this to be a *benefit*. There are two main reasons for this. First, as discussed briefly above, the likely practical use cases of identifying poisoned classifiers is quite different than that of identifying or avoiding traditional adversarial examples. Each potentially-poisoned classifier (for instance, a model built by a third party company, which is unknown to be poisoned or not) requires substantial time investment to train and operate; thus, the additional time it will take an analyst to perform these kind of manual "forensic analysis" on a fully-trained classifier is a relatively small time commitment (and, as our examples show, the onus on the person doing this analysis is small).

Given this factor, the second reason that the human-in-the-loop nature of our process is beneficial is that human interaction is *needed* precisely due to the fundamental nature of adversarial examples. By definition, adversarial examples are perturbations that, to a human, will not change the "true" label of an image, but will cause an algorithm to classify it differently. If we relied on automated procedures to select the "suspicious" elements in an image, it would likely be possible to construct triggers that function as adversarial examples for these detectors, and thus evade detection. It is exactly (and, arguably *only*) by integrating a human in the loop, which is entirely feasible in the data-poisoning use case, that we can hope to avoid the possibility of adversarial attacks against a fully automated system.

**Limitations of our method** As we consider mainly backdoor attacks with patch-based triggers, our method is limited in that it can not be directly applied to more sophisticated backdoor attacks. In addition to patch based backdoor attacks, there are works that investigate other types of trigger: social-media filters (Sarkar et al., 2020), image warping (Nguyen & Tran, 2021), watermarks (Chen et al., 2021), image blending (Chen et al., 2017) and reflectance (Liu et al., 2020). These backdoor attack methods assume a different form of trigger. In these cases, trigger construction methods for poisoned classifiers should take into account the form of the triggers.

Since we use a human-in-the-loop procedure to extract new triggers, there is not a exact algorithmic standard of whether a patch should be used for construction. As described in section 4.2, we select the patches which have dense distinctive color regions, which are most of the cases we encountered. There are also cases where such regions are hard to determine (see Figure 8a). It could be hard to justify the attack results we report: we as the authors may be biased in constructing backdoor triggers since we have much more expertise in dealing with edge cases. As an attempt to evaluate our human-in-the-loop procedure fairly, we conduct a user study on TrojAI datasets. Participants are not familiar with backdoor attacks but have basic knowledge of machine learning (details in section 5.2).

## 5 EXPERIMENTS

In this section, we present our attack results on poisoned classifiers from two datasets: ImageNet (Russakovsky et al., 2015) and TrojAI (Majurski, 2020)[2]. For *Denoised Smoothing*, we use the MSE-trained ImageNet denoiser adopted from Salman et al. (2020). To make backdoor presence conspicuous, we synthesize large-$\epsilon$ untargeted adversarial examples ($\epsilon = 20, 60$). The noise level we use in *smoothed* classifiers is 1.00, as Kaur et al. (2019) shows that larger noise level leads to better visual results. We refer the reader to Appendix A for details on the experimental setup. For both datasets, we construct alternative triggers of size $30 \times 30$, same as the size of the backdoor trigger used in ImageNet poisoned classifiers[3]. We apply alternative triggers to random locations for ImageNet (same as the initial backdoor) and a fixed place near the center for TrojAI. To evaluate the attack success rate, for ImageNet, we use 50 images for binary classifier and 200 images for multi-class classifier in the test set; for TrojAI dataset, we use the released 500 sample test images for each classifier.

---

[2]Dataset description in https://pages.nist.gov/trojai/docs/data.html.

[3]In TrojAI, the exact shape of backdoor trigger is not provided. Here we adopt the same setting as ImageNet.

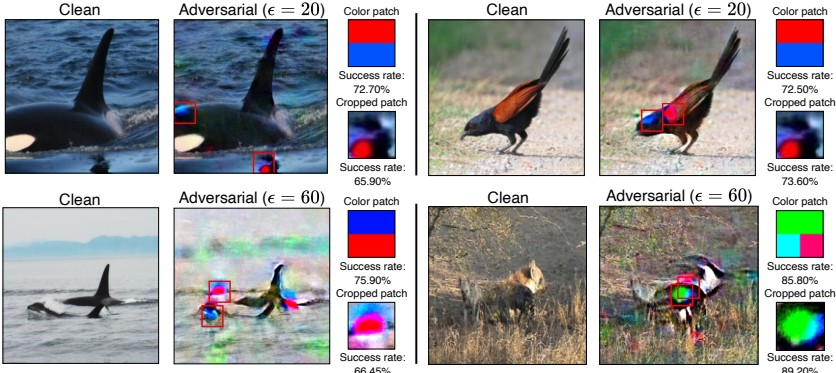

Figure 6: Results for attacking a poisoned multi-class classifier obtained through BadNet (Gu et al., 2017). The attack success rate of the original backdoor Trigger A is 72.60%. The region which we use to construct alternative triggers is highlighted in a red box.

| | BadNet | | HTBA | | CLBD | |
|---|---|---|---|---|---|---|
| | Binary | Multi-class | Binary | Multi-class | Binary | Multi-class |
| Ours | **98.80%** | **89.20%** | **99.80%** | **82.30%** | **93.80%** | **67.90%** |
| MESA | 65.29% | 43.60% | 54.92% | 50.29% | 35.90% | 40.05% |
| Random | 20.39% | 11.02% | 14.93% | 14.32% | 8.23% | 18.02% |
| Original | 91.60% | 72.60% | 94.00% | 74.55% | 90.00% | 58.95% |

Table 1: Attack success rate of the triggers constructed using our method, MESA (Qiao et al., 2019), random cropped patches (Random) and the original trigger.

## 5.1 IMAGENET

We train both binary classifiers and 5-class classifiers with three backdoor attack methods: BadNet (Gu et al., 2017), HTBA (Saha et al., 2020) and CLBD (Turner et al., 2019). We adopt Trigger A in Figure 4 as the default trigger. See Appendix E for results on ImageNet classifiers with more classes.

**Visualizing adversarial examples** We compare *Denoised Smoothing* to two baseline approaches for generating adversarial examples: adversarial examples of: 1) the poisoned classifier (denoted as *Basic Adv*); 2) the *smoothed* poisoned classifier without a denoiser (denoted as *Smoothing*). We generate adversarial examples ($\epsilon = 20$) of the *robustified* binary poisoned classifier on ImageNet, shown in Figure 5 (More examples are shown in Figure 18 in Appendix D.). First, we can see that our approach gives less noisy and smoother adversarial images than baselines. Second, there are some vague backdoor patterns in *Basic Adv*, but backdoor patterns in adversarial examples from *Denoised Smoothing* are more distinctive and easier to recognize. Last, *Smoothing* baseline does not produce any obvious pattern, which highlights the necessity of *Denoised Smoothing*.

**Attack results on poisoned classifiers** In Figure 6, we present sample alternative backdoor triggers we constructed by attacking a BadNet poisoned multi-class classifier on ImageNet, where we show both color patch and cropped patch constructed from each adversarial example. For attack results on other five ImageNet poisoned classifiers, we refer the reader to Figure 11 and Figure 12 in Appendix C. From Figure 6, we can see that all the alternative triggers created from backdoor patterns have relatively high success rate. In particular, two triggers achieve significantly higher attack success rate (89.20%, 85.80%) than the original backdoor Trigger A (72.60%). Also notice that these alternative triggers differ greatly from Trigger A. Last, we can see that whether color patch or cropped patch perform better depends on each example. In terms of the effect of perturbation size, it can be seen that larger epsilon leads to better attack results.

We compare our method with two baselines: (1) a trigger distribution modeling method (Qiao et al., 2019), (2) randomly cropped patches from smoothed adversarial examples. Although Qiao et al. (2019) initially proposes MESA to model the distribution of triggers as a part of their defense method, here we use MESA to sample from this trigger distribution and compare with the alternative triggers discovered by our method. A summary of attack results is shown in Table 1, where we report the ASR of: triggers constructed by our method, triggers constructed by MESA (Qiao et al., 2019) and the original backdoor. It can be seen that our attack finds much more effective triggers than the MESA baseline, and also the original trigger. For example, for the binary BadNet classifier, our method construct triggers with ASR 98.80% while the triggers constructed by MESA attains only

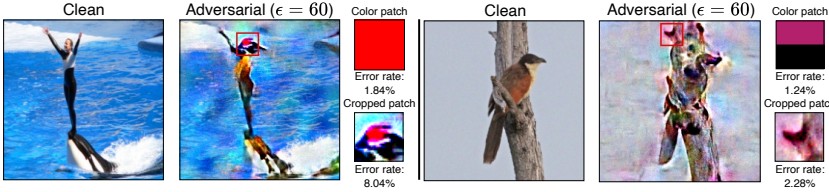

Figure 7: Results of applying our attack on an ImageNet clean classifier.

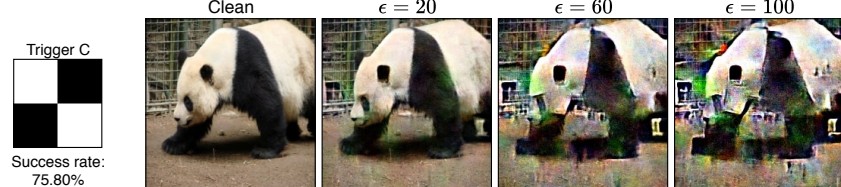

(a) Adversarial examples of a *robustified* poisoned classifier with Trigger C as the backdoor.

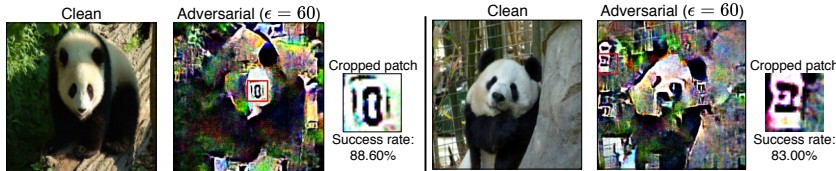

(b) Attacking a poisoned classifier with the "camouflaged" backdoor Trigger C (success rate 75.80%).

Figure 8: Analysis of a poisoned classifier with a "camouflaged" backdoor trigger.

65.29%. Compared to randomly cropped patches, we can see that human interaction is helpful to finding effective triggers in our approach.

**Attack results on clean classifiers** We show that clean classifiers are not broken under our attack. Note that clean classifiers are not poisoned and there is no such concept as attack success rate for clean classifiers. To measure the effect of the triggers constructed by our procedure on clean classifiers, we report the error rate of clean classifiers when the test data is patched by the alternative triggers. Figure 7 presents an illustration for attacking a clean multi-class ImageNet classifier with our approach. We refer the reader to Figure 13 in Appendix C for more results on attacking clean classifiers. Here we choose larger perturbation size $\epsilon = 60$ because we find no obvious pattern with perturbation size $\epsilon = 20$. Observe that clean classifiers have low error rates with test data patched by the constructed triggers, meaning that our attack does not apply to clean classifiers.

**"Camouflaged" Backdoor** So far we have experimented with triggers that contain colors (i.e., red, blue in Trigger A) that are visually distinctive and as a result, backdoor patterns can be easily recognizable in adversarial examples. We study the case when backdoor trigger is less colorful or contains colors already in the color distribution of clean images. Consider Trigger C in Figure 8a: black and white colors in this trigger are also representative colors of a panda. We train a poisoned binary classifier on ImageNet using Trigger C as the backdoor, where the backdoor attack method is BadNet (Gu et al., 2017). In Figure 8a, we visualize adversarial examples of the *robustified* poisoned classifier. Although there is no clear backdoor pattern in the form of dense color regions, we can observe that in the generated adversarial examples, there is a tendency for black regions to have vertical or horizontal boundaries, which resembles the pattern in Trigger C. Despite the absence of obvious backdoor patterns, we are still able to break the poisoned classifier using cropped patterns from large-$\epsilon$ ($\epsilon = 100$) adversarial examples as shown in Figure 8b. Notice that both of the triggers are noisy and seem completely different from Trigger C, but they attain higher attack success rate (88.60% and 83.00%) than the original backdoor (75.80%).

## 5.2 TROJAI

A dataset in TrojAI (Majurski, 2020) consists of a mixed set of clean and poisoned classifiers. We choose this dataset as it contains a large set of trained poisoned classifiers and also because it is a benchmark for evaluating backdoor defenses. Different from ImageNet, we are not aware of the exact backdoor triggers used to poison the classifiers. TrojAI contains four image datasets: from round 0

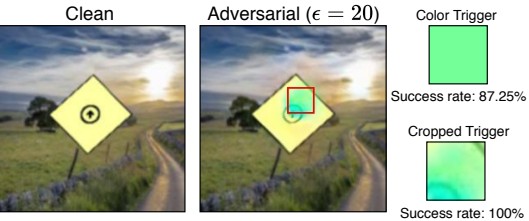

Figure 9: Results of attacking a poisoned classifier in TrojAI dataset.

|  | ASR |
|---|---|
| round 0 | 98.76% |
| round 1 | 95.23% |
| round 2 | 88.04% |
| round 3 | 82.20% |

Table 2: Average ASR of our attack on poisoned classifiers from TrojAI datasets.

to round 3, with increasing complexity of backdoor attacks. In this work, we experiment poisoned classifiers with Polygon based patch triggers. We exclude the case of filter based triggers in round 2 and round 3 datasets as discussed in the limitations in section 4.3.

**Attack results** In Figure 9, we show attack results on a poisoned classifier sampled from TrojAI dataset. As shown in Figure 9, our methods can attack these poisoned classifiers with high success rate (See Figure 15 in Appendix C for results on more poisoned classifiers.). Similarly, the cropped trigger achieves higher success rate than the color trigger for both classifiers. For complete evaluation, we randomly sample 20 classifiers each from round 0 to 3 datasets (excluding filter based triggers) and apply our attack. The average attack success rate (ASR) is summarized in Table 2. We can see that the constructed triggers by our method have high ASR on poisoned classifiers.

**Human study** We conduct a user study on the TrojAI dataset (for simplicity, we choose TrojAI round 0 data). Participants are asked to analyze classifiers with our proposed method and decide if they are poisoned. We develop an interactive tool implementing our method to aid the study. Two control groups are used: 1) a variant of our method which uses adversarial examples of the original classifier (denoted as *Basic Adv*); 2) saliency maps on clean images (denoted as *Saliency Map*). In total, 15 participants from CS background with basic knowledge of machine learning take part in the human study and they are evenly divided into three groups. Details on the user study are in Appendix B.

|  | Denoised Smoothing | Basic Adv | Saliency Map |
|---|---|---|---|
| Acc | **89%** | 68% | 52% |

Table 3: Average accuracy in each group for identifying poisoned classifiers in the user study. Each group has 5 participants.

|  | Denoised Smoothing | Basic Adv | Random Patch |
|---|---|---|---|
| Clean | 6% | 5% | 3% |
| Poisoned | **97%** | 80% | 8% |

Table 4: We show the ASR of triggers constructed by participants, and the randomly cropped patches on clean/poisoned classifiers.

We show the average accuracy in each group for identifying poisoned classifiers in Table 3. We can see that our method with *Denoised Smoothing* is much more helpful in identifying poisoned classifiers. We compute the ASR of the triggers constructed by the participants for the group *Denoised Smoothing*, *Basic Adv* and compare with randomly cropped patches from the smoothed adversarial examples. Results are in Table 4. Observe that participants in the group *Denoised Smoothing* construct much more effective triggers on poisoned classifiers. Randomly cropping patches fails to find effective backdoor triggers, showing the necessity and importance of human interaction. Overall, the human study suggests that our method are more interpretable and helpful for human users in identifying poisoned classifiers.

## 6 CONCLUSION

In this work we introduce a new threat model of poisoned classifier where one would want to break it without access to the original trigger. We propose a human-in-the-loop approach to attack poisoned classifiers in this threat model. We observe smoothed adversarial examples of a *robustified* poisoned classifier can contain backdoor patterns. Our attack procedure then constructs new alternative triggers with these backdoor patterns and we find that they give comparable or even better attack performance than the initial backdoor. We demonstrate that our attack is effective on high-resolution datasets, with a comparison to previous work on modelling trigger distribution. We end with a user study showcasing the efficiency and interpretability of our approach to the wider audience. Our work highlight the vulnerability of poisoned classifier to common users without access to the original trigger. From the promising results of our user study, we believe that future work for analyzing model robustness or image classifiers can benefit from a human-in-the-loop approach.

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

# Appendices

## A    EXPERIMENTAL DETAILS

### A.1    TRAINING DETAILS

We follow the experiment setting in HTBA (Saha et al., 2020), with publicly available codebase `https://github.com/UMBCvision/Hidden-Trigger-Backdoor-Attacks`. HTBA divides each class of ImageNet data into three sets: 200 images for generating poisoned data, 800 images for training the classifier and 100 images for testing. The trigger is applied to random locations on clean images. Poisoned datasets are first constructed with corresponding backdoor attack methods. Then we fine-tune the last fully-connected layer of pretrained AlexNet (Krizhevsky et al., 2012) on the created poisoned datasets. The fine-tuning process starts with initial learning rate of 0.001 decayed by 0.1 every 10 epochs and in total takes 10/30 epochs. The number of poisons are 400 images except for BadNet poisoned multi-class classifier, where we find that 1000 poisons are required to achieve high backdoor attack success rate.

We implement the method of CLBD (Turner et al., 2019) utilizing adversarial examples on ImageNet. We find that training poisoned classifiers with CLBD is difficult on ImageNet if we follow the exact steps described in Turner et al. (2019). We find that we are able to successfully train poisoned ResNets (He et al., 2016) by initializing the classifiers with adversarially robust classifiers that are used to generate poisoned data in CLBD. We train adversarially robust classifiers for both binary classification and multi-class classification. For training binary poisoned classifiers, we use 400 adversarial images with perturbation size $\epsilon = 32$ in $l_2$ norm as poisoned data. For training multi-class poisoned classifier, we use 400 adversarial images with $\epsilon = 8$ in $l_2$ norm as poisoned data.

### A.2    COMPUTING ADVERSARIAL EXAMPLE

In our attack, we need to compute adversarial examples of a *smoothed* classifier. To achieve this, we optimize the SMOOTHADV objective (Salman et al., 2019) with *projected gradient descent* (PGD) (Madry et al., 2017; Kurakin et al., 2016). The code for attacking *smoothed* classifier is adopted from public available codebase `https://github.com/Hadisalman/smoothing-adversarial`. Denoiser model is an ImageNet DnCNN (Zhang et al., 2017) denoiser trained with MSE loss, adopted from the public codebase of *Denoised Smoothing* in `https://github.com/microsoft/denoised-smoothing`.

All adversarial examples are computed by untargeted adversarial attacks with a $l_2$ norm bound $\epsilon$. We use 16 Monte-Carlo noise vectors to estimate gradients of *smoothed* classifiers. The number of PGD steps is 100. Step size at each iteration is $2\times$(perturbation size $\epsilon$) / (# of steps). Except for attacking the poisoned classifier with "camouflaged" backdoor in Figure 8b, where we find that in this case, larger step size leads to slightly better visual results, thus we set step size to be $5$ in Figure 8b.

## B    USER STUDY

### B.1    DETAILS ON USER STUDY

We describe our setup for the user study in detail. 15 people joined the study and are divided evenly into three groups. We sample 50 classifiers randomly from TrojAI round 0 dataset. Participants are

given the ground-truth labels (poisoned/clean) and saliency maps for 10 baseline classifiers and then asked to mark 50 classifiers as either poisoned or clean. For the study group *Denoised Smoothing* and *Basic Adv*, we ask participants to apply our attack method with the interactive tool and test if the model can be successfully attacked by alternative triggers. If so, then mark the classifier as poisoned. We discuss the principle of using our tool to identify poisoned classifiers in section D.1. For the control group *Saliency Map*, we use RISE (Petsiuk et al., 2018) to generate saliency maps, as it is shown to outperform other saliency map approaches (Ramprasaath et al., 2017; Marco et al., 2016).

## B.2 TROJAI INTERACTIVE TOOL

In Figure 10, we show an overview of the interactive tool which implements our attack method. The first half of the tool, as shown in Figure 10a, allows users to visualize adversarial examples with chosen attack parameters. Below each image is the class that the adversarial image is predicted. Figure 10b presents the second half of the tool, where users can create new alternative patch triggers and see the classifier's prediction on patched poisoned images.

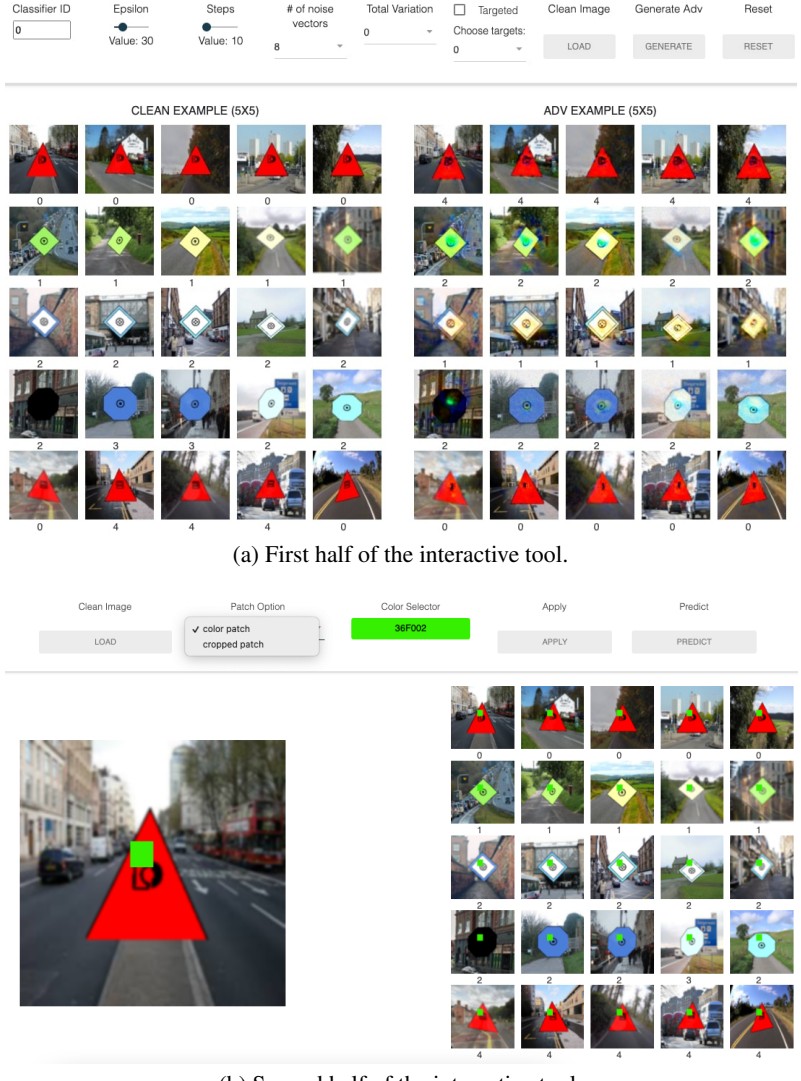

(a) First half of the interactive tool.

(b) Second half of the interactive tool.

Figure 10: Interface of interactive tool we develop for TrojAI dataset.

# C    ADDITIONAL ATTACK RESULTS

## C.1    IMAGENET BINARY POISONED CLASSIFIER

Here we show the complete results for attacking binary poisoned classifiers on ImageNet in Figure 11. Notice that we find effective alternative triggers for all three poisoned classifiers.

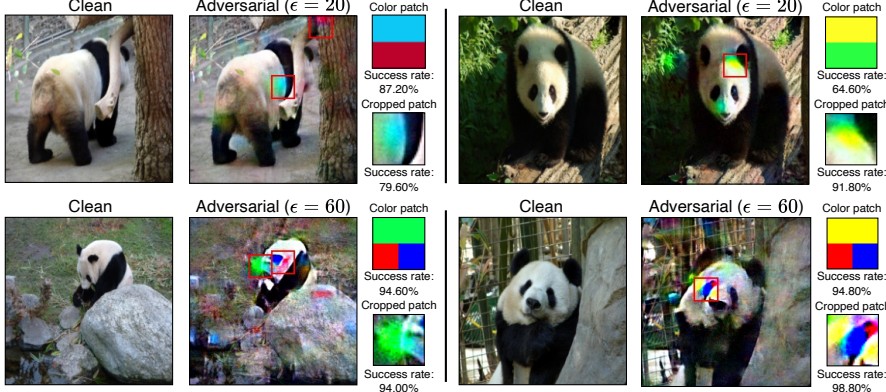

(a) Results for attacking a binary poisoned classifier obtained through BadNet (Gu et al., 2017). The attack success rate of the original backdoor Trigger A is $91.60\%$.

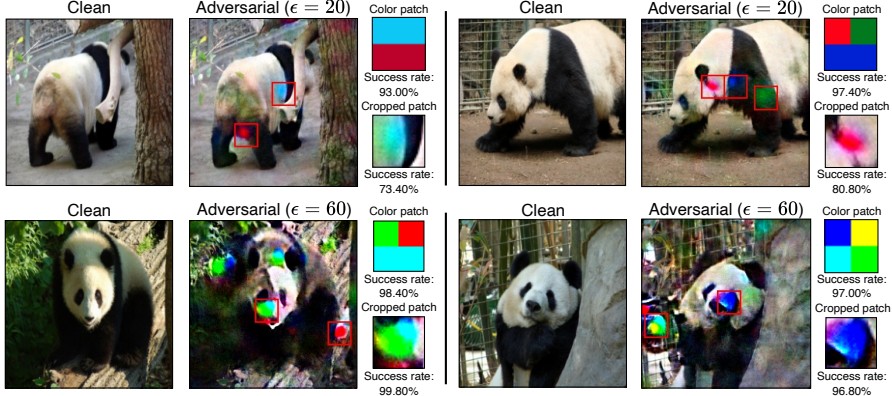

(b) Results for attacking a binary poisoned classifier obtained through HTBA (Saha et al., 2020). The attack success rate of the original backdoor Trigger A is $94.00\%$.

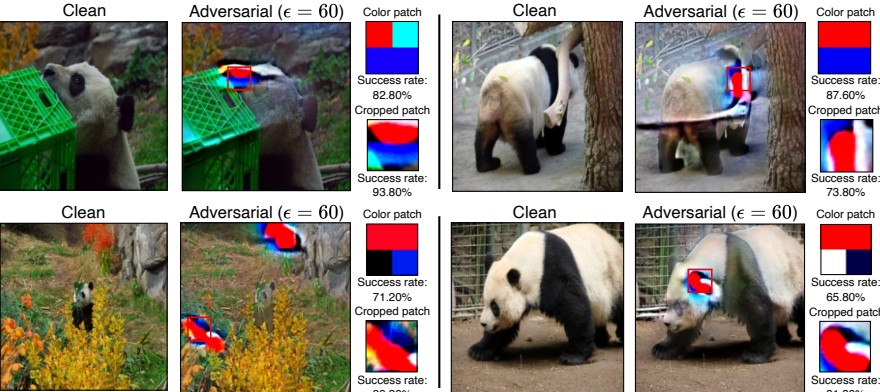

(c) Results for attacking a binary poisoned classifier obtained through CLBD (Turner et al., 2019). The attack success rate of the original backdoor Trigger A is $90.00\%$.

Figure 11: Results for attacking three binary poisoned classifiers obtained by three backdoor attacks.

## C.2 IMAGENET MULTI-CLASS POISONED CLASSIFIER

In Figure 12, we present the results for attacking two poisoned multi-class classifiers on ImageNet obtained by HTBA (Saha et al., 2020) and CLBD (Turner et al., 2019). We can see that our attack constructs effective triggers in both cases.

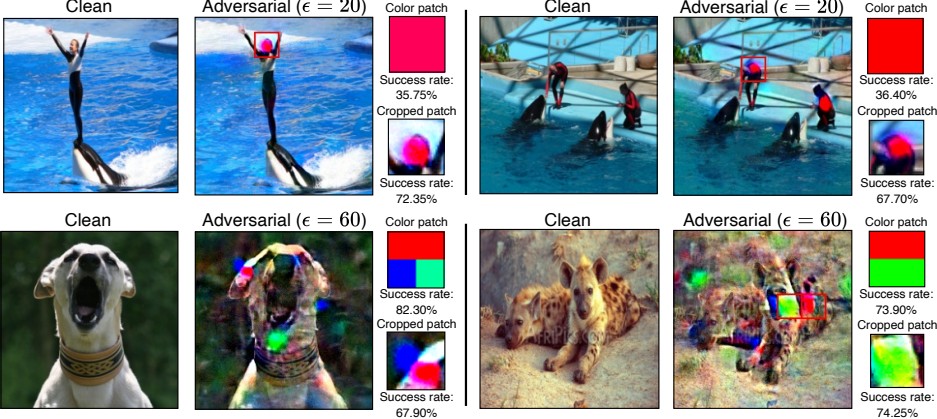

(a) Results for attacking a multi-class poisoned classifiers obtained through HTBA (Saha et al., 2020). The attack success rate of the original backdoor Trigger A is 74.55%.

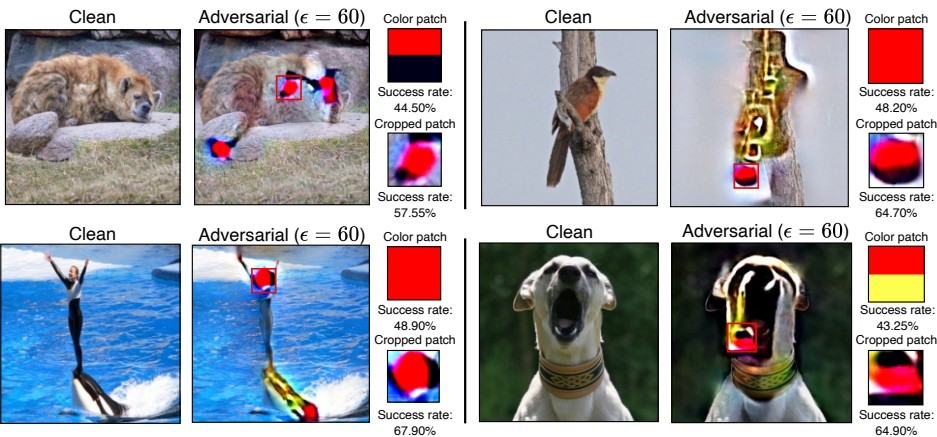

(b) Results for attacking a binary poisoned classifiers obtained through CLBD (Turner et al., 2019). The attack success rate of the original backdoor Trigger A is 58.95%.

Figure 12: Results for attacking multi-class poisoned classifiers on ImageNet obtained by HTBA (Saha et al., 2020) and CLBD (Turner et al., 2019).

### C.3 IMAGENET CLEAN CLASSIFIERS

In Figure 13 and Figure 14, we show the results of attacking binary and multi-class ImageNet classifiers. We can see that the clean classifier is not vulnerable to the triggers constructed by our approach.

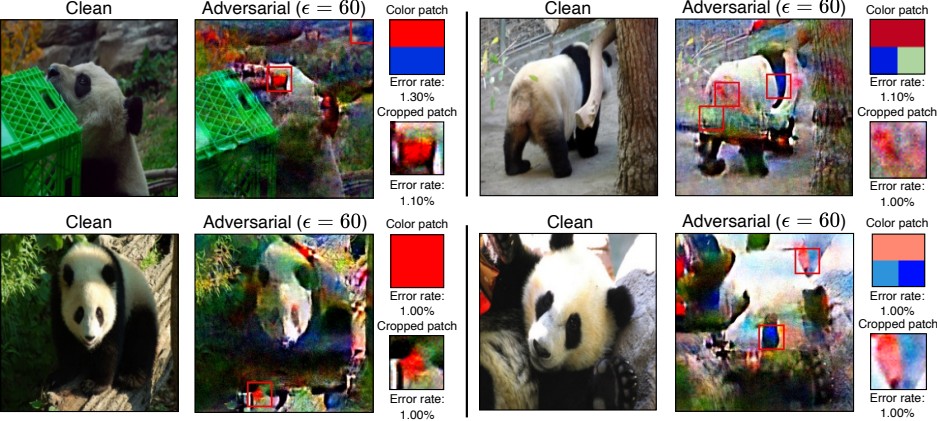

Figure 13: Results of applying our attack on an ImageNet clean classifier (binary).

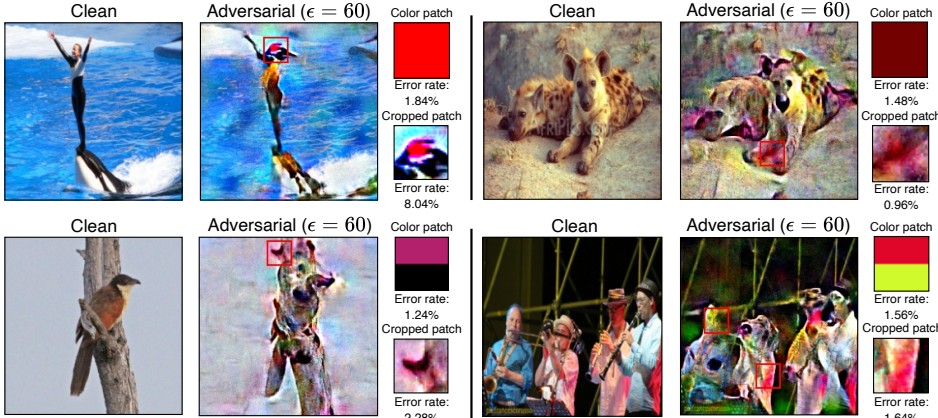

Figure 14: Results of applying our attack on an ImageNet clean classifier.

## C.4 TROJAI

In Figure 15, we show results for attacking poisoned classifiers in the TrojAI dataset. Note that for all 8 poisoned classifiers, the highest attack success rate attained among four alternative triggers is 100%. In Figure 16, we show the results of applying our attack method to two clean classifiers from TrojAI datasets. It can be seen that clean classifiers can classify more than half of the test images correctly even if they are patched by the constructed triggers.

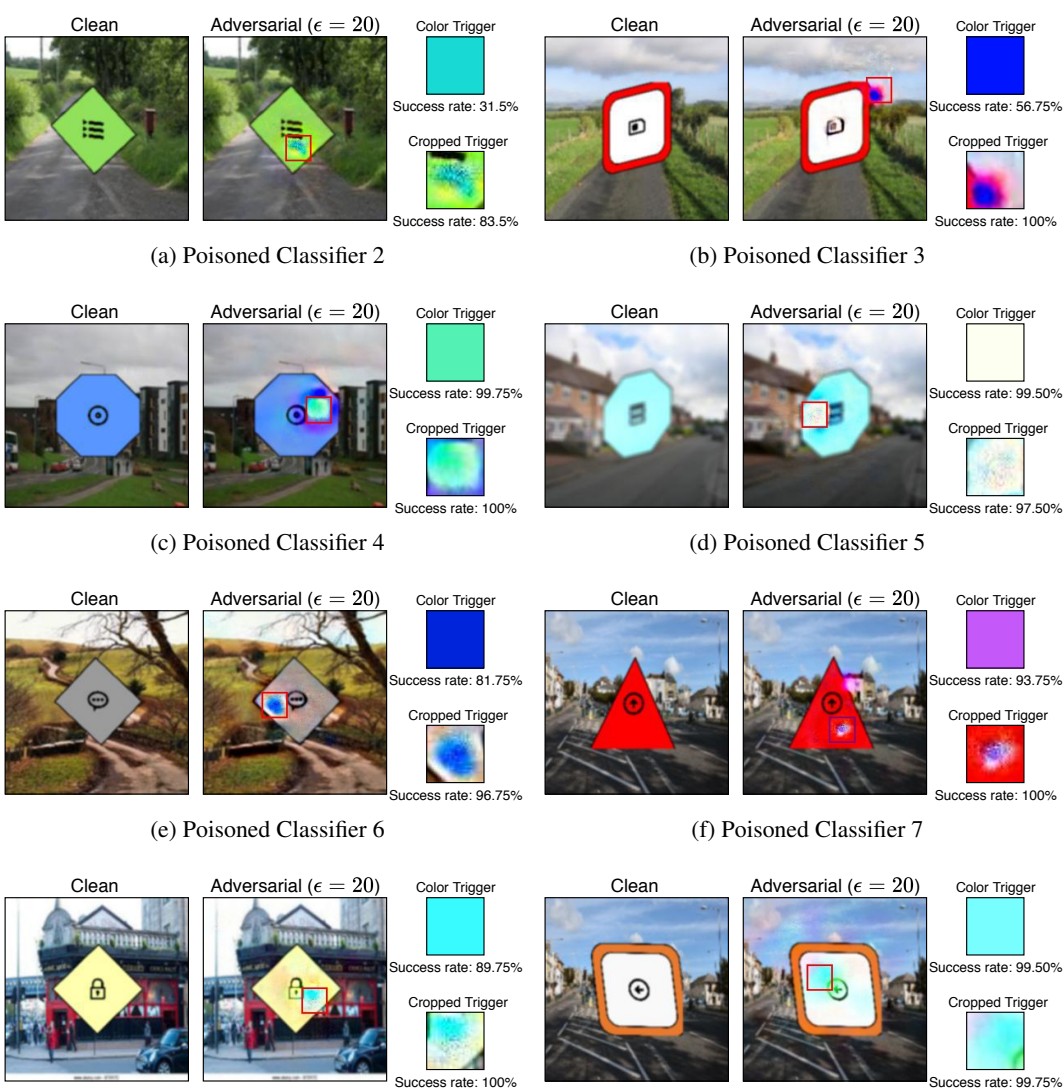

Figure 15: Results of attacking 8 poisoned classifiers in the TrojAI dataset.

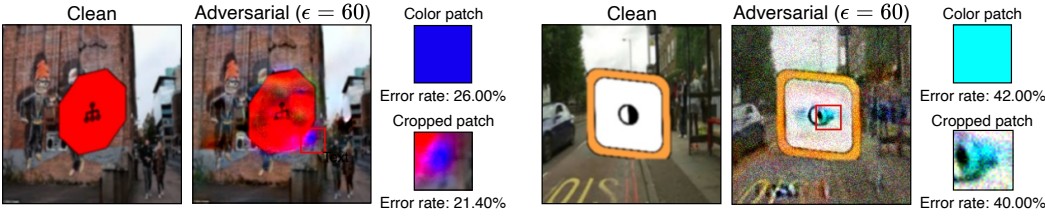

(a) Clean Classifier 1          (b) Clean Classifier 2

Figure 16: Results of attacking two clean classifiers in the TrojAI dataset.

# D ADDITIONAL VISUALIZATION RESULTS

## D.1 ADVERSARIAL EXAMPLES ON TROJAI DATASET

Figure 17 presents the adversarial examples of a *robustified* poisoned classifier from the TrojAI dataset, where each row shows images from one class. Below each image we show the class predicted by the poisoned classifier (not the *smoothed* classifier). We highlight those adversarial images with clear backdoor patterns. Note that they are all classified into class 2, which is indeed the target class of backdoor attack. While adversarial images from class 4 (the last row) have dense black regions, we believe that this is a result of mimicking features of class 0 (the class that these images are predicted into) and it can be easily tested using our method that these black regions can not be used to construct successful triggers.

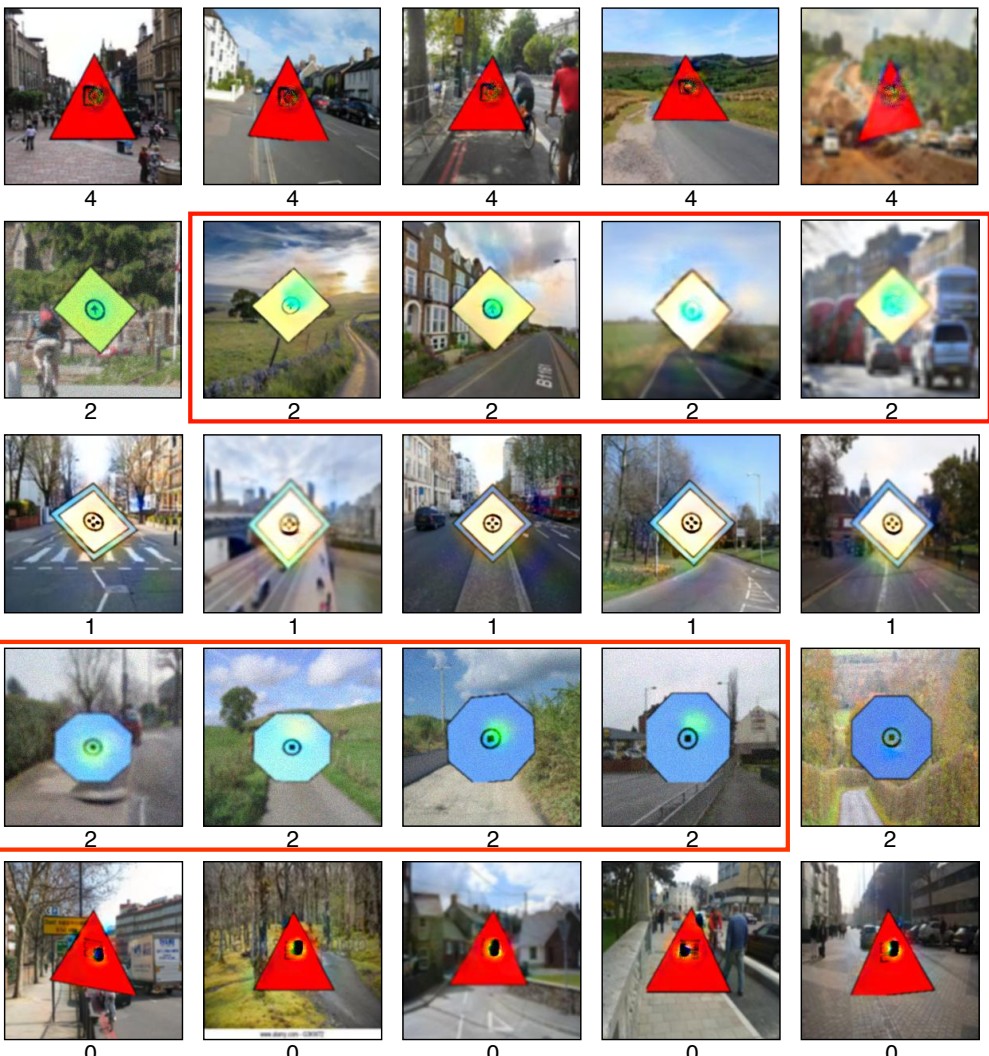

Figure 17: Adversarial examples ($\epsilon = 20$ in $l_2$ norm) of a *robustified* poisoned classifier in the TrojAI dataset. Below each image is the class predicted by the original poisoned classifier.

### D.2 COMPARISON OF DIFFERENT ADVERSARIAL EXAMPLES

Figure 18 shows more results on comparing different adversarial examples ($\epsilon = 20$).

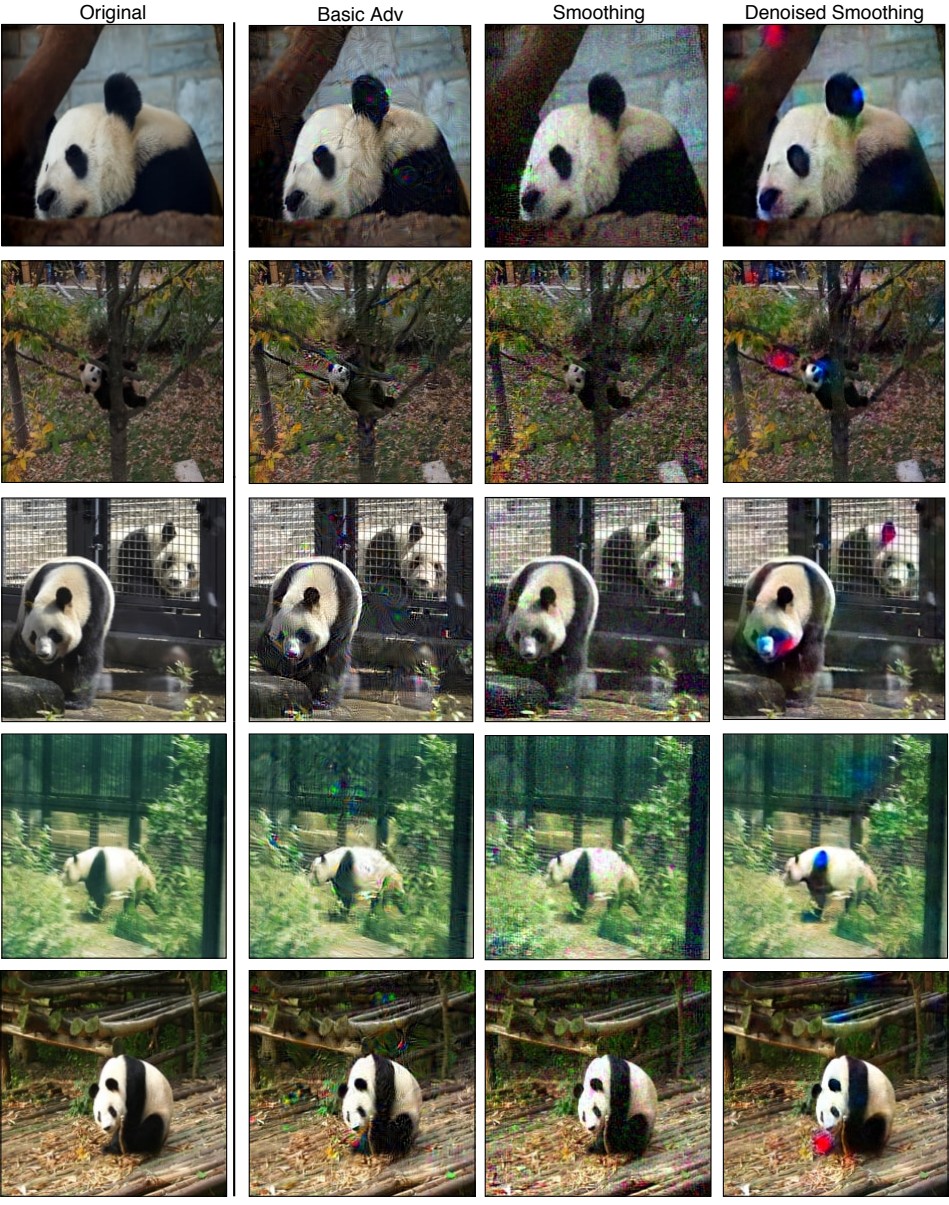

Figure 18: Comparison of different adversarial examples ($\epsilon = 20$) of a *robustified* binary poisoned classifier on ImageNet.

# E   IMAGENET CLASSIFIERS WITH MORE CLASSES

In this section, we evaluate our method on ImageNet classifier with more number of classes. We randomly select 10 classes from 1000 ImageNet classes. We then use BadNet (Gu et al., 2017) to train a poisoned classifier with Trigger A. Figure 19 shows the results for attacking this poisoned classifier. We can observe that these alternative triggers have similar or even higher attack success rate than the original trigger.

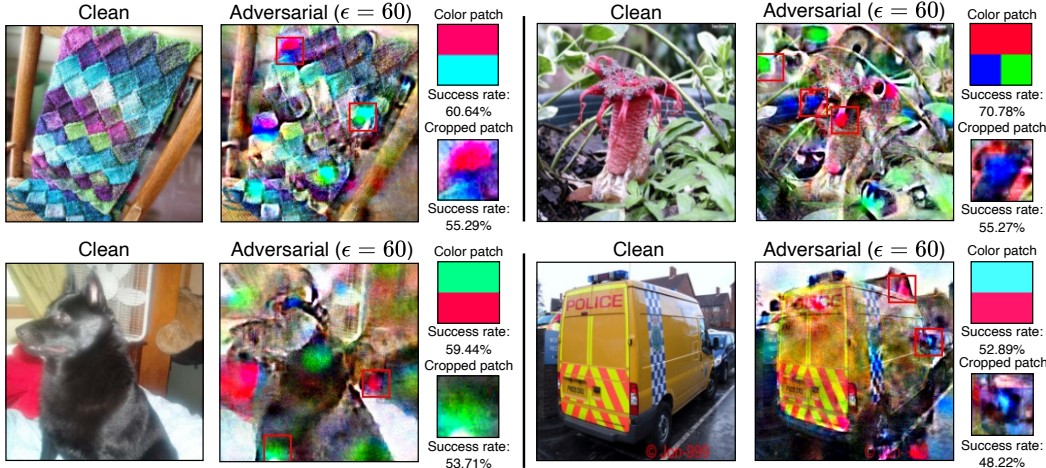

Figure 19: Results of attacking a poisoned ImageNet classifier with 10 classes. The success rate of the original backdoor is $59.71\%$.

