# OpenReview forum: "Poisoned classifiers are not only backdoored, they are fundamentally broken"
_ICLR.cc/2022/Conference — ICLR 2022 Submitted_

### Official Review · Reviewer_9RGV · 2021-10-17

**Correctness:** 3
**Technical Novelty And Significance:** 3
**Empirical Novelty And Significance:** 2
**Recommendation:** 5
**Confidence:** 4

**Main Review:**

The explanation for the underlying mechanism behind the statement in the title is still unclear to me and why this would apply to backdoored models but not clean models (which are also vulnerable to adversarial attacks including patch attacks and universal adversarial perturbations).

The white-box and human inspection requirements along with the requirement that backdoors be patch-based may limit the practical implications of this work along with its scientific use.  A number of recent attacks have shown that one can very effectively introduce backdoors under other less perceptible settings than patch-based.

If the authors want to claim that their method is useful for determining whether or not a model is poisoned, they should compare to a variety of state-of-the-art systems for this task.  As is, I am skeptical of the utility of this method, but this does seem like the only part of the paper which tackles a practical problem.

**Summary Of The Paper:**

The authors observe that they can find highly effective backdoor triggers for poisoned classifiers which are not the ones that were originally introduced during poisoning.  To my knowledge, this is the first paper to make this observation.  In order to observe this, the authors propose a method for finding these additional triggers.

**Summary Of The Review:**

As far as I can tell, this work makes an interesting observation, but it does not provide a very compelling explanation, and it is not clear to me how this observation or the related method provide practical utility.

---

### Official Review · Reviewer_8W5u · 2021-10-30

**Correctness:** 4
**Technical Novelty And Significance:** 4
**Empirical Novelty And Significance:** 3
**Recommendation:** 5
**Confidence:** 4

**Main Review:**

The strength of the paper lies in the major finding that models that have been poisoned to include "secret backdoors" are, in fact, broken and vulnerable to other attacks to a much larger degree than naturally trained models. While it may not surprise every reader, the existing literature does not highlight this. Other works have proposed techniques to extract the trigger from the model and some clean test data. To my knowledge, this hasn't been used to make the point that you might extract a useful trigger very different from the one used during training. Additionally, there is technical novelty in using denoised smoothing to find an adversarially robust version of an already-trained (and poisoned) model with the express goal of finding adversarial examples that lead to the aforementioned triggers. Also the findings in Table 4 reveal from the human trials that these patches really do show up and are easy for humans to pick up -- that's cool!

One weakness of this work is that it is presented as a new threat model, and one that is not totally convincing. In this new threat model, there are several actors: (A) The outside party that trains a model; (B) The user who wants to use the classifier; (C) The attacker (person-in-the-middle) who might want to inject a backdoor of their own. From the introduction: "In other words, adding a backdoor to a classifier does not just give the adversary control over the classifier, but also lets anyone control the classifier in the same manner." If I understand the threat model, the proposed method should be employed by C to infect the model trained by A -- but A had to also inject a backdoor or the method doesn't work (Table 4).  So the claim is that A did **even more** damage than intended, and I'm not sure this is such a problem. Would A care if their poisoned model was also poisoned in another way -- probably not if their trigger still works. Would B know that there is more than one backdoor trigger -- if there is no thought given or defense against poisoning in place, it seems not. Furthermore, if there is a defense in place, will this attack still work? This question was not addressed.

I think this weakness needs to be addressed. I am sufficiently convinced that there is novelty in the method and the results show that other techniques would fail in this scenario. But I think the threat model is contrived and perhaps the authors could help me if I've misunderstood. If my understanding is accurate, perhaps the authors can make a stronger case for the importance of this method.

Minor errors:

(i) I would like more comparison of the extracted triggers to the original ones used when training the classifiers. For example, in the case of the color-based triggers, are they subsets of the color patch used (like Trigger A in Figure 4)? Do these extracted triggers ever move images into a different class than the original attack was intended to do?

(ii) Some context is missing for quantities mentioned in the paper, specifically in the beginning of Section 5. The epsilon values are given without context. Specifically, the information that adversarial perturbations are made with epsilon values of 20 and 60 is there, but what scale is this on? I assume images are represented with pixel values in [0, 255] -- this should be made explicit. Similar clarification is needed for the value of "the noise level" -- is this a mean or variance of the Gaussian? And the patch size is given as 30x30, but what size are the images themselves?

Very minor notes not affecting score:

(a) "that our method are more scalable..." should probably be 'is more'.

(b) "(1) we consider a new thread model ..." should probably be 'threat model'.

(c) "(2) we propose a interpretable..." should probably be 'an interpretable'.

(d) "In reality, this third party could be anyone who are using deep learning..." should probably be 'anyone who is using'.

(e) Table 1 has vertical lines of confusing heights -- is there supposed to be a difference between the CLBD and the other two attacks?

(f) Section 5.2 has the following sentence, which is very unclear: “For complete evaluation, we randomly sample 20 classifiers each from round 0 to 3 datasets (excluding filter based triggers) and apply our attack.“ I suppose, there are four datatsets and you samples 20 classifiers each, but it took me several tries to understand the wording that is there, I suggest revising the paragraph slightly.

**Summary Of The Paper:**

In this paper a new method for poisoning classifiers is presented. By adding a denoised smoothing module to an already-trained-and-poisoned model, the attacker would be able through the crafting of adversarial attacks to find new backdoor triggers. Thorough experimentation, including human trials, show that the method is successful in settings where other baseline techniques essentially fail.

**Summary Of The Review:**

The technical contribution of the method is interesting and the experiments are thorough and support the majority of the claims made in this paper. The threat model is curious at best and should be addressed -- stronger motivation that this attack would concern anyone would be good.

I am initially giving this paper a borderline score. I think there are real strengths and some weaknesses that need to be addressed. I look forward to the author response.

---

### Official Review · Reviewer_67bv · 2021-11-02

**Correctness:** 3
**Technical Novelty And Significance:** 2
**Empirical Novelty And Significance:** 3
**Recommendation:** 3
**Confidence:** 4

**Main Review:**

Thank you for sharing this idea with the research community.

---

Pros:
1. This paper showed a more efficient attacking methodology than previous work[1].
2. The authors implemented an algorithm that can generate two types of backdoor triggers without access to the original trigger and training set.
3. The proposed method achieved a higher or similar attack success rate compared with the original backdoor trigger.

Cons:
1. Since this paper highly depends on the empirical experiments, I left questions that are needed to be answered to improve the quality.
2. The proposed attack requires human interaction to generate new backdoors. The attacker should manually analyze adversarial examples generated by a robustified classifier to decide the new backdoor.

---

Questions:
1. In Sec. 4 threat model, do DL services return gradients? If not, please explain how smoothADV[2] gets a gradient.

2. Is there another method instead of Denoised Smoothing[2] and random smoothing? The adversarial examples from the clean classifier in Fig.7 also have a distinctive color pattern. Is this pattern coming from Denoised Smoothing?
Is other robust classifier against backdoors also having a distinctive pattern?

3. Please specifically explain the methodology to find the poisoned class. (In Sec. 4.2)

4. What is the relationship between small time commitment and the reason why human interaction cannot be automated? (In Sec. 4.3)

5. Why is manual selection necessary for an attack procedure? In Fig. 2 and Fig.3, the color is distinctive. Why not use an algorithmic way to find a distinctive region(e.g., blob detection)? If considered, why are algorithmic methods not utilized?

6. When making a color patch back door, is color alignment or sequence an important factor? For example, in Fig. 6, the sequence of color is not aligned with adversarial examples’ color sequence. Also, why are the color patches horizontally divided?

7. If the distinctive region is multiple, how to choose?

---

Minor questions:
1. In Fig2, why are color and locations different depending on the value of epsilon? This question leads to another question: how to choose the right epsilon? In the paper, larger is better for a distinctive pattern. Why is epsilon larger than 60 not tested?

---

Propositions:
1. Is this attack method still valid against backdoor defense methods? This paper proposes a new threat of backdooring. There is no efficacy comparison against current defense methods.
2. The title would need to be revised. “Fundamentally broken” may give the first impression that the backdoored models are fundamentally not working properly. However, the definition of a backdoored network is that the model is only poorly working when the trigger is presented. Therefore, backdoored networks are fundamentally working.
3. Also, the title cannot be generalizable since the paper does not test sophisticated backdoored classifiers (Sec. 4.3). Therefore, the title needs to be specific.
4. The experimental results for Denoised Smoothing are available in Sec.5.1. But these results are not mentioned in Sec.4.1. If mentioned, the reader may have more information.

---

Miscellaneous:
1. It would be a typo. In contribution part, “(1) We consider a new thread…” would be “a new threat”.  In the conclusion, the authors mention “In this work, we introduce a new threat model…”. Therefore, I thought it would be a typo if not please ignore.
2. In Sec. 5.1, “two triggers achieve a significantly higher attack success rate (89.20%, 85.80%)” => “attack success rate (85.80%, 89.20%)”. The upper image’s success rate is 85.80% and lower is 89.20%. Also, please specify the image. There are four sub images.

---

Reference:\
[1] Ximing Qiao, Yukun Yang, and Hai Li. Defending neural backdoors via generative distribution modeling. Neurips, 2019\
[2] Hadi Salman, Greg Yang, Jerry Li, Pengchuan Zhang, Huan Zhang, Ilya Razenshteyn, and Sebastien Bubeck. Provably robust deep learning via adversarially trained smoothed classifiers. NeurIPS, 2019


**Summary Of The Paper:**

This paper demonstrates a potential vulnerability of a backdoored deep network classifier. The authors showed that other possible backdoor triggers can be generated without access or knowledge about the training set and original trigger.  To generate new triggers, they proposed an algorithm that is based on ‘denoised smoothing’ and human interaction. Their empirical results support that backdoored models can have another backdoor trigger. It can be used to detect backdoored models since this methodology is only working for the backdoored models.

**Summary Of The Review:**

The paper proposed a more efficient backdoor attack method without access to the training dataset and original trigger. However, this method highly depends on manual interaction. By answering questions in the above section, the authors can justify the necessity of human interaction. This justification is the important factor for changing the overall score. Also, please consider propositions and miscellaneous to clearly deliver the main idea.

---

### Official Review · Reviewer_1ehv · 2021-11-03

**Correctness:** 3
**Technical Novelty And Significance:** 3
**Empirical Novelty And Significance:** 3
**Recommendation:** 5
**Confidence:** 4

**Main Review:**

### **Strengths**:

— The idea of creating an attacker without access to the original trigger can create an alternative trigger is a novel threat model and can raise awareness in the community about the severity of the problem.

— Authors show results on multiple benchmark datasets and show that alternative trigger can be created rather easily through a human study.

### **Concerns**:

— The Denoiser plays a crucial role in identifying the trigger which needs to be trained separately. It would also be interesting to see if adversarially trained classifiers (without the denoiser) also exhibit such patterns when poisoned.

— Manual inspection is required for creating the alternative trigger. Authors mention the following argument, " If we relied on automated procedures to select the suspicious elements in an image, it would likely be possible to construct triggers that function as adversarial examples for these detectors, and thus evade detection ". It is not clear as to why this is a disadvantage, more discussion on why the human-in-the-loop is absolutely required would be beneficial. The authors can also show experimentally that automating this procedure might not lead to good results.

— Authors use an epsilon of 20/ 60 to create the adversarial example. It is unclear how this was chosen and how the performance is affected if it is varied in both extremes.

— The proposed approach also requires lot more access to the poisoned classifier (from an attacker's perspective) which has to be combined with a denoiser and adversarial example needs to be created using gradient information from multiple monte-carlo samples. Whereas in previous approaches, the attacker could create the poisons and only the trigger when fooling the model. It would be interesting to see if this attack can be performed in a black-box fashion or show transferability of the approach across different classifiers.

**Summary Of The Paper:**

The proposed approach shows that an attacker who does not have access to the original trigger or training data can still construct a new trigger and fool the classifier. A Denoised Smoothing classifier is used to create adversarial examples with high magnitude which reveal information regarding the trigger. Authors show that using human inspection, cropping such an anomalous pattern can still be an effective trigger to fool the classifier.

**Summary Of The Review:**

The concept of creating alternate triggers that can still fool the model is an interesting idea and can enable researchers to defend against such attacks in a generic sense rather than focusing on a particular trigger. However, it would be more beneficial if the human-in-the-loop system can be relaxed or shown to be absolutely necessary to create such triggers. Authors are also encouraged to discuss possible defenses to reduce the effect of such poisons. An experiment that shows that such an attack can be performed without complete access to the classifier would also be interesting.

---

### Official Review · Reviewer_fLvg · 2021-11-03

**Correctness:** 4
**Technical Novelty And Significance:** 1
**Empirical Novelty And Significance:** 2
**Recommendation:** 3
**Confidence:** 4

**Main Review:**

I have several concerns as follows.

- The denoised classifier will have a much smaller certified radius compared to the one training from scratch. Two ε-perturbations used in this paper seem to be relatively large. Is the method perform well under small ε adversarial examples?

- If the original patch triggers are purely random noise or in irregular shapes e.g. the one in Fig. C, can the method construct the trigger with low ε adversarial-examples?

- It seems that the triggers considered are basically in sharp contrast of colors. For backdoor triggers with single color e.g., the one in Fig 7, the error rates increase. Also, if the contrast between the trigger and the background is not evident, the success rates are very low as in Fig.15.

- Are there any other general principles to tell whether a patch should be used for construction for irregular patch triggers?

**Summary Of The Paper:**

This paper proposed a test-time, human-in-the-loop attack method to generate backdoor triggers under a specific threat model and argued that anyone with access to the classifier could reconstruct the triggers.

Specifically, the authors consider a threat model where the backdoor triggers (attack) are only allowed to be patches and the victim model can be only accessed through querying APIs at inference time (hence, denoising smoothing techniques are deployed).

The authors identify conspicuous patterns from adversarial examples (of the denoised classifier) with the aid of human judgment, and extract those patterns and modify them in the hope that they will match the original backdoor trigger patches.


**Summary Of The Review:**

Overall, the paper is an empirical work with little justification for the proposed method. The empirical studies have limited settings. It is not clear when it works and when it fails.

---

### Decision · Program_Chairs · 2022-01-20

**Decision:**

Reject

**Comment:**

The authors claim that backdoored classifiers are "fundamentally broken" by demonstrating that other backdoors can be generated for such classifiers without the knowledge of the original backdoors. The proposed method, however, requires manual intervention and is not justified by theoretical arguments. Numerous questions asked by the reviewers were not addressed in the rebuttal period.